# Epidemiology, Physiology and Clinical Approach to Sleepiness at the Wheel in OSA Patients: A Narrative Review

**DOI:** 10.3390/jcm11133691

**Published:** 2022-06-27

**Authors:** Maria R. Bonsignore, Carolina Lombardi, Simone Lombardo, Francesco Fanfulla

**Affiliations:** 1PROMISE Department, University of Palermo, 90127 Palermo, Italy; 2Sleep Clinic, Division of Respiratory Medicine, Ospedali Riuniti Villa Sofia-Cervello, 90146 Palermo, Italy; ing.s.lombardo@gmail.com; 3Institute for Biomedical Research and Innovation (IRIB), National Research Council (CNR), 90146 Palermo, Italy; 4Sleep Disorders Center, Department of Cardiology, San Luca Hospital, Istituto Auxologico Italiano, IRCCS, 20145 Milan, Italy; c.lombardi@auxologico.it; 5Department of Medicine and Surgery, University of Milano-Bicocca, 20126 Milan, Italy; 6Respiratory Function and Sleep Unit, Maugeri Clinical and Scientific Institute of Pavia and Montescano, 27100 Pavia, Italy; francesco.fanfulla@icsmaugeri.it

**Keywords:** maintenance of wakefulness test, Epworth Sleepiness Scale, motor vehicle accidents, commercial drivers, driving license

## Abstract

Sleepiness at the wheel (SW) is recognized as an important factor contributing to road traffic accidents, since up to 30 percent of fatal accidents have been attributed to SW. Sleepiness-related motor vehicle accidents may occur both from falling asleep while driving and from behavior impairment attributable to sleepiness. SW can be caused by various sleep disorders but also by behavioral factors such as sleep deprivation, shift work and non-restorative sleep, as well as chronic disease or the treatment with drugs that negatively affect the level of vigilance. An association between obstructive sleep apnea (OSA) and motor vehicle accidents has been found, with an increasing risk in OSA patients up to sevenfold in comparison to the general population. Regular treatment with continuous positive airway pressure (CPAP) relieves excessive daytime sleepiness and reduces the crash risk. Open questions still remain about the physiological and clinical determinants of SW in OSA patients: the severity of OSA in terms of the frequency of respiratory events (apnea hypopnea index, AHI) or hypoxic load, the severity of daytime sleepiness, concomitant chronic sleep deprivation, comorbidities, the presence of depressive symptoms or chronic fatigue. Herein, we provide a review addressing the epidemiological, physiological and clinical aspects of SW, with a particular focus on the methods to recognize those patients at risk of SW.

## 1. Introduction

Obstructive sleep apnea (OSA) is a highly prevalent disease characterized by upper airway occlusion during sleep, intermittent hypoxemia, and sleep fragmentation [1]. Patients with OSA are at an increased risk for cardiometabolic disease and car accidents. The most widely used treatment for OSA is the application of continuous positive airway pressure (CPAP) during sleep, which re-establishes airway patency and prevents the occurrence of upper airway collapse and its functional consequences [1]. However, CPAP is not always accepted or tolerated by many patients, and the compliance with treatment is highly variable. The use of CPAP for at least 4 h on 70% of the nights is considered the threshold for good compliance, but regular and extended CPAP use beyond this threshold is associated with a larger improvement in OSA symptoms, including excessive daytime sleepiness [2].

The role of OSA in increasing sleepiness at the wheel (SW) and the associated risk of driving and occupational accidents has been recognized since the early clinical research studies, as confirmed by several meta-analyses [3,4,5]. OSA doubles the risk of driving accidents and near-miss accidents, but effective CPAP treatment with good compliance was found to normalize the risk [6,7,8].

Since the literature on the topic of driving risk in OSA is extensive, the reader is referred to some reviews for a summary of the previous literature [9,10,11]. A recent systematic review has examined the issue of driving risk in OSA in great detail [10]. The current narrative review aims at providing an overview with a focus on recent updates regarding the epidemiology, the pathophysiology and practical suggestions to evaluate SW in OSA patients.

## 2. Epidemiology of Driving Accidents in the General Population and in OSA Patients

Excessive daytime sleepiness (EDS) is a non-specific symptom potentially caused by several factors which may be associated with an increased risk of traffic accidents. Most commonly, EDS is the consequence of insufficient sleep, and a recent study comparing the amount of sleep reported by French drivers between 1996 and 2011 found that the sleep time of drivers decreased over 15 years, and this finding was associated with an increased prevalence of SW [12]. Fatigue or sleep-related accidents have been known for a long time to be frequent causes of traffic accidents in the general population [13,14,15], even though accidents attributable to sleepiness show an estimated rate between 10 and 30% [13,16,17,18]. Moreover, car crashes related to falling asleep often cause death and severe injury [14]. The death of the driver occurred in 11.4% of sleepiness-related accidents, in contrast with 5.6% of accidents unrelated to sleep [17]. Sleepiness-related motor vehicle accidents (MVA) may result from falling asleep while driving and behavior impairment attributable to sleepiness [18].

Besides OSA, other known causes of EDS are: (1) neurological diseases, such as narcolepsy, idiopathic hypersomnia or Parkinson disease [19,20]; (2) psychiatric disease, especially depression [21]; (3) old age [22]; (4) the use of hypnotics [23]; and (5) diabetes [24]. Sleepiness at the wheel should be specifically investigated, since the risk for car accidents is high in subjects reporting episodes of severe SW [25] and in patients with OSA [26]. Although elderly subjects often report sleepiness, the risk for a poor driving performance associated with sleepiness is lower in old adults compared to young adults, possibly because the former tend to avoid what they perceive as dangerous situations—for example, driving at night [27]. Conversely, risky driving behavior was associated with an increased risk for accidents [28].

The overall picture shows several areas of uncertainty with regard to the factors possibly predictive of SW and sleepiness-related accidents. Nevertheless, the majority of research studies worldwide reported an increased driving risk among those with untreated OSA [10], and meta-analyses further confirmed this finding [2,4,5]. Untreated OSA increases the risk of motor vehicle accidents by 1.5 to 2.5 times compared to the general population, and the risk might be especially high in the excessively sleepy OSA clinical phenotype [29]. The role of OSA in driving accidents is further confirmed by the decreased/normalized accident rate after the initiation of CPAP treatment, evaluated as actual accidents, near-miss accidents or driving simulator performance [7,8].

More recently, in a nationwide cohort study performed in Denmark, Udholm et al., found a prevalence of motor vehicle accidents (MVA) in OSA patients that was 1.4%, higher than the prevalence in the reference population (0.98%) after the adjustment for age, sex, socioeconomic status and co-morbidities. The hazard ratio for MVA in patients with OSA was 1.29 (IC 1.18–1.39), while the incidence rate ratio was 1.3 (1.2–1.42). Interestingly, CPAP therapy determined a statistically significant risk reduction for MVA in comparison to OSA patients that were not treated, with a hazard ratio of 0.82 (0.67–1.02) and an incidence rate ratio of 0.75 (0.6–0.91) [30]. Another recent study was performed in older subjects to assess the effect of sleep apnea on driving behavior. OSA severity, measured by AHI, increased the likelihood of an adverse driving event, with a 1.25 times increase in the odds of an event for each eight-point increase in the AHI [31]. This is the first study that reported a “dose-effect” association between OSA severity and the risk of MVA.

Several studies have tried to identify the predictors of car accidents in OSA patients [10]. The apnea-hypopnea index (AHI), as a marker of OSA severity, was reported to be associated with the occurrence of driving accidents by some studies [32,33], while sleepiness was the principal factor in other studies [26,34,35,36,37]. Besides the occurrence of SW, near-miss car accidents in OSA patients could also provide interesting information. SW was reported by 41.3% of OSA patients, but in some cases, it was not associated with excessive daytime sleepiness, as evaluated by the ESS score [37]. SW was predicted by the ESS score, depression and the level of exposure, i.e., the mileage per year. Near-miss accidents were reported by 22% of patients reporting SW and were associated with ESS, depression, habitual sleep duration and the oxygen desaturation index (ODI) [37]. Compared to AHI, the severity of nocturnal hypoxia seems to be a better marker of daytime sleepiness [37,38,39] or poor performance at psychomotor vigilance tests [40]. The lack of a relationship between AHI and the risk for car accidents could at least partly be explained by the finding that OSA is not always associated with daytime sleepiness, as shown by recent studies on clinical OSA phenotypes [29,41,42,43]. Excessive sleepiness is common in obesity, independent of coexisting OSA [10].

Some metabolic biomarkers may be related with the occurrence of sleepiness in OSA patients. Among them, increased interleukin-6 (IL-6) has been found during sleep restriction in normal subjects [44] and in OSA patients with objectively documented sleepiness, i.e., a short sleep latency at multiple sleep latency tests, whereas no correlation was shown between IL-6 and subjective sleepiness, as assessed by the Epworth or Stanford Sleepiness scale [45]. In untreated OSA patients, subjective sleepiness was associated with a poor performance for the Psychomotor Vigilance Task (PVT) test but not with IL-6 levels [46]. In women with OSA, increased IL-6 was independently associated with ESS, low physical activity and depression [47], but it was unaffected by CPAP treatment for 12 weeks [48]. Similar findings were reported in a randomized clinical trial in patients with coronary artery disease and non-sleepy OSA after long-term CPAP treatment [49].

C-reactive protein (CRP) is an inflammatory biomarker associated with untreated OSA [50] which decreased shortly after the initiation of CPAP treatment [51]. CRP has recently been assessed as a risk factor for OSA in four prospective studies in population cohorts. Although increased CRP levels at baseline increased the risk for incident OSA, the association was attenuated after the adjustment for BMI, indicating a major role of obesity. The effect of CRP on OSA risk was larger in younger and nonobese subjects [52]. There is currently little evidence for an association of CRP levels with sleepiness [48]. Overall, these results confirm the complexity of the interactions between IL-6, inflammatory markers, OSA, obesity and sleepiness and suggest a limited usefulness of inflammatory biomarkers in predicting EDS in OSA.

Finally, as already discussed, insufficient sleep is a major cause of car accidents in the general population [12], and comorbidities might also contribute, as is the case in depression [21] and diabetes [24].

The results of epidemiological studies on driving risk in OSA should be critically evaluated due to several sources of variability. Firstly, the methods used to assess sleepiness varied, with the majority of studies using the Epworth Sleepiness Scale (ESS) which measures subjective sleepiness in eight situations [53]. The repeatability of the results of ESS has been questioned [54,55], and a low ESS score may be falsely reported despite the occurrence of SW [56]. Secondly, the occurrence of OSA in drivers has been determined as OSA risk by using questionnaires or by the objective documentation of sleep disordered breathing by polysomnography or cardiorespiratory polygraphy [10]. Thirdly, the population under study is a major source of variability, since differences in the results exist between studies in the general population, patients with suspected or diagnosed OSA and non-commercial versus commercial drivers [10]. Fourthly, some studies reported only car accidents that were objectively documented, while other studies derived the risk of car accidents from tests assessing sleepiness, such as the multiple sleep latency test (MSLT) or the maintenance of wakefulness test (MWT) [10]. The use of driving simulators might seem best suited to specifically address driving risk, but the current evidence suggests a limited prediction of sleepiness-related driving risk [57].

Long-haul truck drivers represent a high-risk population due to their high driving distance/year, i.e., increased exposure, frequent sleep debt, and a high prevalence of OSA [11,58,59]. The study by Burks and coworkers in truck drivers in the United States showed that drivers with OSA who were nonadherent to CPAP treatment had a fivefold risk of serious preventable crashes, whereas those adherent to CPAP treatment showed a crash rate similar to the controls [60]. The economic implications are that the motor vehicle-accident-related costs associated with the lack of effective treatment are very high—much higher than those of effective CPAP treatment [60,61]. Legislation on commercial drivers is a very delicate issue, with the interest for public safety on one side and privacy regulations and economic issues on the other side [62,63]. The European Union Directive 2014/85/EU specifically considered the problem of OSA for issuing or renewing a driving license and took into account different safety profiles for commercial and non-commercial drivers [64], but EU Member States have adopted individual measures, and the problem of driving safety in OSA patients is currently heterogeneously addressed throughout Europe.

## 3. Pathophysiology of Sleepiness at the Wheel in OSA Patients

The identification of risk factors also depends on the methods used to assess sleepiness. Since ESS scores are subjective and show many limitations, the maintenance of wakefulness test (MWT) is currently considered to be the most effective tool to study sleep propensity with regard to SW, especially if the 40 min protocol is used rather than the usual 20 min protocol [65,66,67,68]. However, in addition to sleep latency, which is the main result of the MWT, very short episodes of sleep during MWT could be more sensitive markers of the risk to experience car accidents by reflecting drowsiness, i.e., the intermediate state between wakefulness and established sleep. Drowsiness preceding sleep is associated with a lack of control and may be a crucial determinant of car accidents [69,70,71,72]. The automatic identification of microsleep episodes may be the first step to develop a new clinical test to identify risky drivers [73]. Age is another important risk factor for car accidents, particularly in patients with OSA. Very recently, Doherty et al., found that higher sleep apnea severity was associated with a higher incidence of adverse driving behavior, such as hard acceleration, braking and speeding. These findings were also observed in cognitively unimpaired old individuals [31].

Another physiological aspect that should be considered is the reduced performance of OSA patients in multi-tasking tests. Driving can be considered a complex and dynamic task that requires the simultaneous performance of intrinsic sub-tasks, such as vehicle control and traffic management, as well as other activities such as radio listening, navigator control, smartphone use, etc. Huang et al., reported that OSA patients showed a reduced performance for divided attention driving simulator trials in comparison to normal age-matched controls [74]. Mazza et al., evaluated the driving performance in a road safety platform, a more natural driving performance test, in a group of OSA patients before and after CPAP treatment [75]. The driving task consisted of avoiding an aquatic obstacle during three real driving conditions: (1) simple condition; (2) distraction condition; (3) anticipation condition. During the simple driving conditions, the patients showed increasing reaction times and a lengthening of the vehicle stopping distance in comparison with the control group. The distraction condition caused a lengthening in reaction times, particularly in OSA patients. Of interest, CPAP therapy determined a statistically significant reduction in the reaction times in OSA patients in comparison to the baseline, as well as a reduction in stopping distance in the distraction and anticipation driving conditions [75]. On the other hand, the ability to perform a multi-tasking trial can be impaired by sleep restriction, including a chronic partial sleep restriction [76]. In this way, OSA per se and partial sleep restriction may have an additive effect on the reduction in driving performance. Furthermore, Vakulin et al., demonstrated that patients with OSA are more vulnerable than healthy individuals to the effects of alcohol consumption or sleep restriction on driving performance [77].

## 4. Evaluation of Sleepiness at the Wheel in OSA Patients

Although our knowledge on the determinants of SW has greatly increased, the evaluation of the risk for car accidents in OSA patients remains problematic when considering the number of subjects to be tested for driving license issuing or renewal and the lack of simple and fast tests to be used clinically on a large scale [78]. OSA is recognized as a relevant cause of car accidents worldwide, but the identification of subjects at risk is far from being satisfactory. Figure 1 and Figure 2 schematically depict a possible flow chart for the clinical assessment in subjects with a history of previous car accidents and in subjects with suspected or known OSA, respectively [10].

Different EEG-derived parameters have been proposed to discriminate patients at risk of poor daytime cognitive performance. Parek et al., demonstrated, in a group of OSA patients, that a sustained run of inspiratory flow limitation determined a significant increment of K-complexes density, with a reduced slow-wave activity associated with delta frequencies. The reduced delta activity (ΔSWAK) was associated with next-day lapses in vigilance during a 20 min psychomotor vigilance test (PVT) [79]. In another study, the same research group demonstrated that the improvement in PVT lapses during CPAP therapy was associated with an increase in ΔSWAK [80]. Mullins et al., observed that sleep EEG microstructure measures recorded during routine PSG were associated with impaired vigilance in OSA patients after sleep deprivation [81]. Eight OSA patients underwent baseline PSG followed by wakefulness for 40 h with repeated PVT tests and driving sessions on a driving simulator. The authors found that a greater EEG slowing during REM sleep was associated with slower PVT reaction times, more PVT lapses and more crashes during driving simulator trials. Furthermore, the decreased spindle density during NREM sleep was also associated with slower PVT reaction times [81]. The study involved a very small number of subjects, highlighting the complexity of such protocols and their obvious limitation for use in large samples.

There is growing evidence that OSA is associated with regional changes in sleep electroencephalography (EEG) pattern, reduced fMRI-measured brain connectivity or reduced functional interhemispheric connectivity [82,83]. Azabarzin et al., recently explored the functional implications of the reduced interhemispheric sleep depth coherence for motor vehicle crash risk in middle-aged and older individuals with OSA [84]. The OSA patients with the highest degree of sleep depth coherence were those with a lower risk of being in an accident [84].

A high level of clinical skills and responsibility is demanded for clinicians who should evaluate the patient’s fitness to drive. Specific training would also be useful, since a recent survey reported a high degree of variability in physicians’ responses to cases of fitness to drive evaluations [85]. The diagnosis of excessive daytime sleepiness is still based on subjective rather than objective data, and besides the easiest clinical situations—e.g., patients with no previous accidents and no referred daytime sleepiness, patients on CPAP treatment classified as low risk for accidents and patients with clinically evident excessive daytime sleepiness classified as high or very high risk based on ESS—there remains a large and challenging grey area that needs better definition. In this context, a sleep study and MWT remain the best tools to objectively document sleep propensity, evaluated as sleep or microsleep latency at MWT. Psychomotor vigilance tests are promising but far from being standardized for clinical use [10].

## 5. The New Wake-Promoting Agents

Solriamfetol and pitolisant are new drugs that counteract daytime sleepiness. They are available on the market and are mainly used in neurological disorders such as narcolepsy or idiopathic hypersomnia. Both have been successfully tested in CPAP-treated patients with residual sleepiness and in OSA patients refusing CPAP [86,87,88]. Little information on the prevention of car accidents with these drugs is available, but solriamfetol improved driving performance [89]. The practical use of solriamfetol or pitolisant in OSA patients is still uncertain, since the clinical features of OSA patients who would benefit most from their use are unclear, and the treatment for sleepiness without concomitant treatment might have detrimental effects on other consequences of OSA, such as cardiometabolic diseases.

## 6. Conclusions

The role of OSA in increasing the risk for driving accidents is established, and effective CPAP treatment decreases the risk. However, OSA is not the only cause of sleepiness at the wheel, and there are no simple methods to identify the patients at high risk for accidents. From a public health point of view, the risk for driving accidents associated with untreated OSA has been recognized by the EU Commission Directive [64]. Research is exploring new markers of sleepiness, such as the occurrence of microsleep episodes during MWT and polysomnographic indicators of heightened sleep propensity, in order to improve the identification of patients at a high risk. Although new wake-promoting drugs active on daytime sleepiness are available, their clinical indications in OSA patients are still poorly defined.

## Figures and Tables

**Figure 1 jcm-11-03691-f001:**
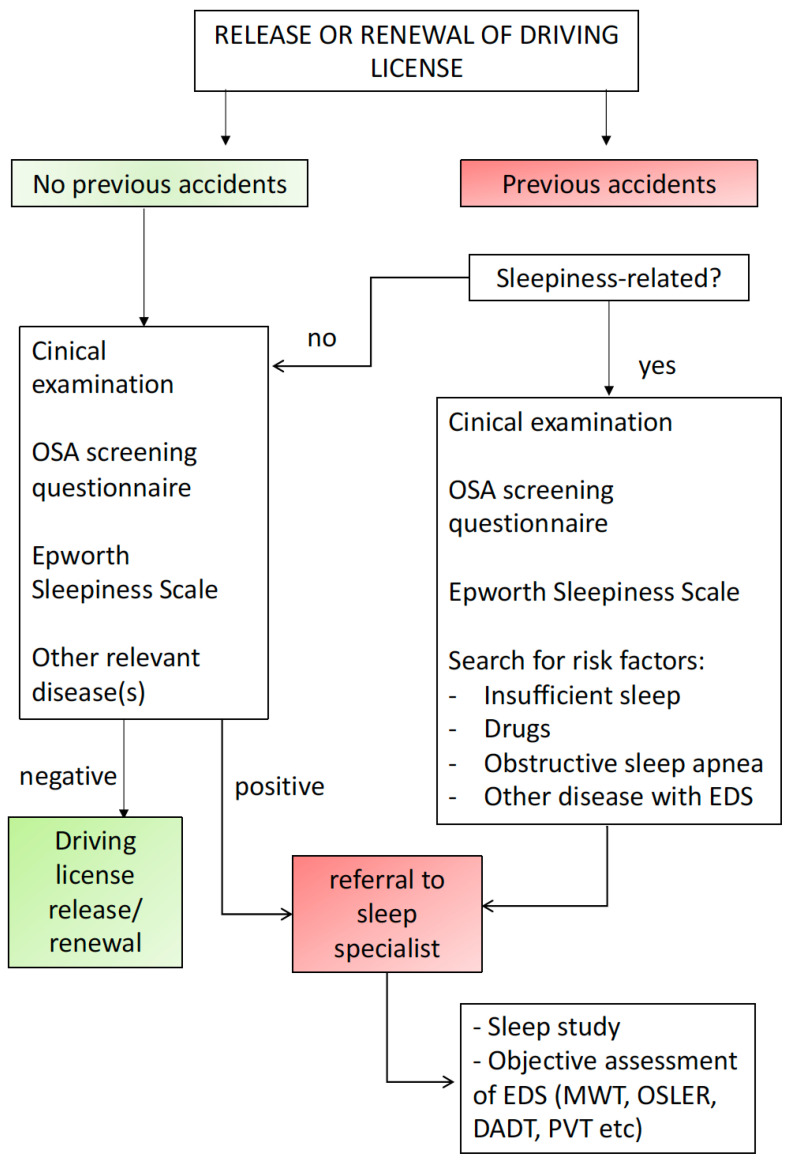
Proposed flowchart to be used for driving license release or renewal. Occurrence of previous sleepiness-related car accidents should be thoroughly investigated. Abbreviations: OSA: Obstructive Sleep Apnea; EDS: Excessive Daytime Sleepiness; MWT: Maintenance of Wakefulness Test; Test; DADT: Divided Attention Driving Task; PVT: Psychomotor Vigilance Test. From [10] with permission.

**Figure 2 jcm-11-03691-f002:**
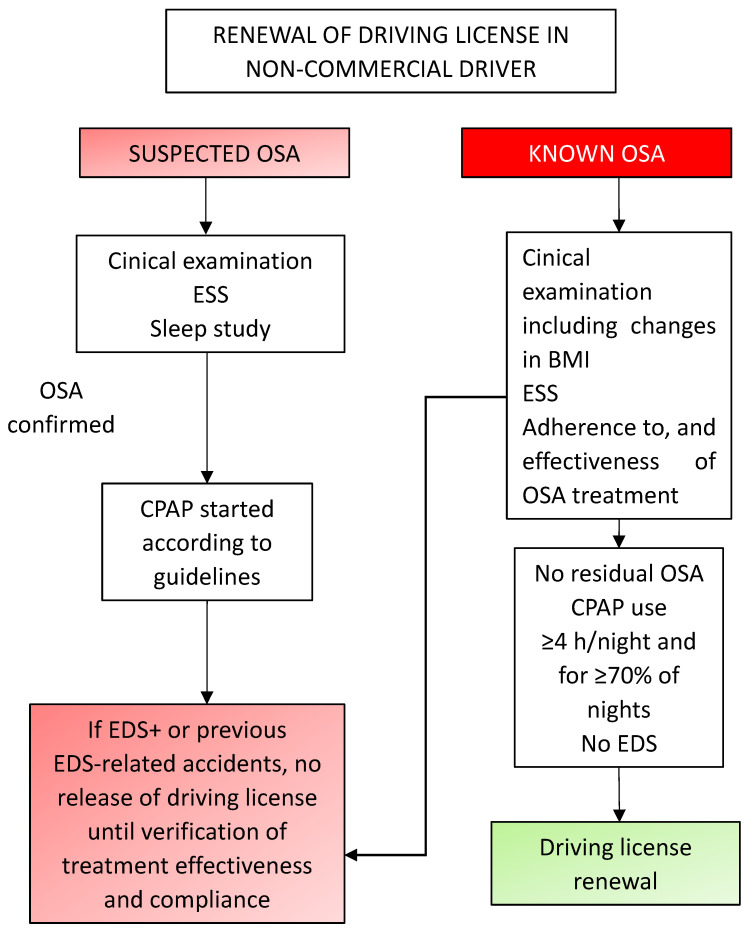
Proposed flowchart to be used for driving license release or renewal. Patients with suspected obstructive sleep apnea (OSA) should be studied and treated whenever the diagnosis of OSA is confirmed. In both newly diagnosed or already known OSA patients, driving licenses should be released or renewed only after the objective documentation of satisfactory compliance with treatment. Abbreviations: ESS: Epworth Sleepiness Scale; BMI: Body Mass Index; CPAP: Continuous Positive Airway Pressure; EDS: Excessive Daytime Sleepiness. Modified from [10] with permission.

## Data Availability

Not applicable.

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
