# Peer review of "Epidemiology, Physiology and Clinical Approach to Sleepiness at the Wheel in OSA Patients: A Narrative Review"

_jcm, 2022, doi:10.3390/jcm11133691_

Round 1

Reviewer 1 Report

Thank you for opportunity to review the paper „ Epidemiology, physiology and clinical approach to sleepiness at the wheel in OSA patients: a narrative review. The manuscript is well written and  flow is good. The manuscript is of interest both clinicians and researchers.

Minor revisions:

  • The abbreviations should be extended both in abstract (i.e. OSA, CPAP, EDS) and figures.
  • 115 authors indicate nocturnal hypoxemia as marker of EDS. However, others metabolic indicators related to EDS should be also discussed i.e. inflammatory markers (CRP etc)
  • Authors propose driving lisence release/renewal if patients CPAP use >4 h/night or >70% of nights. However, effectiveness of CPAP is also important. The residual AHI threshold should be considered.

Author Response

Thank you for your comments

  • The abbreviations should be extended both in abstract (i.e. OSA, CPAP, EDS) and figures.
  • Response: done
  • 115 authors indicate nocturnal hypoxemia as marker of EDS. However, others metabolic indicators related to EDS should be also discussed i.e. inflammatory markers (CRP etc)
  • Response: a new paragraph on IL-6 and CRP has been added (lines 113-134 of the revised manuscript)
  • Authors propose driving lisence release/renewal if patients CPAP use >4 h/night or >70% of nights. However, effectiveness of CPAP is also important. The residual AHI threshold should be considered.
  • Response: Figure 2 has been modified as suggested by the reviwere

Reviewer 2 Report

This narrative review conducted by Bonsignore et al to evaluate sleepiness at the wheel in OSA patients. The topic is valuable and well written, but a few considerations could strengthen the content of this manuscript, as followings:

1.      Although this journal does not require completed methodology for a narrative review, but adding the following information would strengthen this article. Please address the research selection, such as years considered, language, publication status, study design, and databases of coverage. For the narrative, please state limitations and/or quality of research reviewed, and need for future research.

You can follow the checklist at the link:

https://www.elsevier.com/__data/promis_misc/ANDJ%20Narrative%20Review%20Checklist.pdf

2.      For the epidemiology section, I would expect more numbers, such as incidence of SW and in OSA patients, or the correlations between SW, accidences in SW, etcetera. Line 106-122 should be included in epidemiology rather the pathophysiology.

3.      Line 65-79 and 139-161 maybe more suitable in section 4 (evaluation).

4.      Line214-223 discussion two medications for treatment of daytime sleepiness. Management of daytime sleepiness is an importance issue. It seems does not fit for the section, entitled “evaluation of…”

Author Response

This narrative review conducted by Bonsignore et al to evaluate sleepiness at the wheel in OSA patients. The topic is valuable and well written, but a few considerations could strengthen the content of this manuscript, as followings:

Response: Thank you for your positive comments and for the suggestions to move some text around. We hope the manuscript improved in fluency and logical order after revision.

  1. Although this journal does not require completed methodology for a narrative review, but adding the following information would strengthen this article. Please address the research selection, such as years considered, language, publication status, study design, and databases of coverage. For the narrative, please state limitations and/or quality of research reviewed, and need for future research.

You can follow the checklist at the link:

https://www.elsevier.com/__data/promis_misc/ANDJ%20Narrative%20Review%20Checklist.pdf

Response: Since our systematic review was published in 2021, we had agreed with the Editor to provide an concise summary and update of the exisiting literature.

  1. For the epidemiology section, I would expect more numbers, such as incidence of SW and in OSA patients, or the correlations between SW, accidences in SW, etcetera. Line 106-122 should be included in epidemiology rather the pathophysiology.

Response: We have added some numbers on the relevance of SW in the general population and in OSA patients as accidents are concerned. Lines 106-122 were moved to the epidemiology (lines 98-112 of the revised manuscript).

  1. Line 65-79 and 139-161 maybe more suitable in section 4 (evaluation).

Response Lines 65-79 were moved to lines 138-153 of the revised manuscript.

Lines 139-161 were moved to lines 184-204 of the revised manuscript.

  1. Line214-223 discussion two medications for treatment of daytime sleepiness. Management of daytime sleepiness is an importance issue. It seems does not fit for the section, entitled “evaluation of…”

Response: We inserted a new Section 5 entitled: The new wake-promoting agents

Round 2

Reviewer 2 Report

1. Previous comment regarding the search method and checklist is not mentioned in the text. 

2. The references in lines of 113-134 are irrelevant to sleepiness at the wheel in OSA patients, rather only relevant to OSA patients in general. 

Author Response

  1. Previous comment regarding the search method and checklist is not mentioned in the text. Reply: In lines 47-52, the text says: 

    "Since the literature on the topic of driving risk in OSA is extensive, the reader is referred to some reviews for a summary of previous literature [9-11]. A recent systematic review has examined the issue of driving risk in OSA in great detail [10]. The current narrative review aims at providing an overview with a focus on recent updates regarding the epidemiology, the pathophysiology and practical suggestions to evaluate SW in OSA patients."  This was agreed with the Editor upon invitation.

2. The references in lines of 113-134 are irrelevant to sleepiness at the wheel in OSA patients, rather only relevant to OSA patients in general. 

Reply: Your comment is very well taken, but the metabolic markers were added upon request of Reviewer 1. We agree that there is little evidence for any relationship between CRP and SW in OSA, and  modified the text accordingly.
